Identification of hub genes and biological pathways in hepatocellular carcinoma by integrated bioinformatics analysis

Zhao Qian 1
Zhang Yan 1
Shao Shichun 2
Sun Yeqing 554951052@qq.com 2
Lin Zhengkui 2111897652@qq.com 1
1 College of Information Science and Technology, Dalian Martime University , Dalian , Liaoning , China
2 College of Environmental Science and Engineering, Dalian Martime University , Dalian , Liaoning , China
Yang Rendong
Electronic publication date: 2021 Jan 19
Publication date: 2021
Volume: 9
Electronic Location ID: e10594
Received 2020 Jun 8; Accepted 2020 Nov 26
Copyright: ©2021 Zhao et al.
Copyright year: 2021
Copyright holder: Zhao et al.
License: This is an open access article distributed under the terms of the Creative Commons Attribution License, which permits unrestricted use, distribution, reproduction and adaptation in any medium and for any purpose provided that it is properly attributed. For attribution, the original author(s), title, publication source (PeerJ) and either DOI or URL of the article must be cited.
License URL: https://creativecommons.org/licenses/by/4.0/

Keywords: Hepatocellular carcinoma, Gene co-expression network, Biological pathway, Hub gene

Funding: National Science Foundation of China 31770918 Strategic Priority Research Program of the Chinese Academy of Sciences XDA04020202-12 XDA04020412 This work was supported by the National Science Foundation of China (Grant No. 31770918) and the Strategic Priority Research Program of the Chinese Academy of Sciences (Grant No. XDA04020202-12 and XDA04020412). The funders had no role in study design, data collection and analysis, decision to publish, or preparation of the manuscript.

==============================
Background

Hepatocellular carcinoma (HCC), the main type of liver cancer in human, is one of the most prevalent and deadly malignancies in the world. The present study aimed to identify hub genes and key biological pathways by integrated bioinformatics analysis.

Methods

A bioinformatics pipeline based on gene co-expression network (GCN) analysis was built to analyze the gene expression profile of HCC. Firstly, differentially expressed genes (DEGs) were identified and a GCN was constructed with Pearson correlation analysis. Then, the gene modules were identified with 3 different community detection algorithms, and the correlation analysis between gene modules and clinical indicators was performed. Moreover, we used the Search Tool for the Retrieval of Interacting Genes (STRING) database to construct a protein protein interaction (PPI) network of the key gene module, and we identified the hub genes using nine topology analysis algorithms based on this PPI network. Further, we used the Oncomine analysis, survival analysis, GEO data set and random forest algorithm to verify the important roles of hub genes in HCC. Lastly, we explored the methylation changes of hub genes using another GEO data (GSE73003).

Results

Firstly, among the expression profiles, 4,130 up-regulated genes and 471 down-regulated genes were identified. Next, the multi-level algorithm which had the highest modularity divided the GCN into nine gene modules. Also, a key gene module (m1) was identified. The biological processes of GO enrichment of m1 mainly included the processes of mitosis and meiosis and the functions of catalytic and exodeoxyribonuclease activity. Besides, these genes were enriched in the cell cycle and mitotic pathway. Furthermore, we identified 11 hub genes, MCM3, TRMT6, AURKA, CDC20, TOP2A, ECT2, TK1, MCM2, FEN1, NCAPD2 and KPNA2 which played key roles in HCC. The results of multiple verification methods indicated that the 11 hub genes had highly diagnostic efficiencies to distinguish tumors from normal tissues. Lastly, the methylation changes of gene CDC20, TOP2A, TK1, FEN1 in HCC samples had statistical significance (P-value < 0.05).

Conclusion

MCM3, TRMT6, AURKA, CDC20, TOP2A, ECT2, TK1, MCM2, FEN1, NCAPD2 and KPNA2 could be potential biomarkers or therapeutic targets for HCC. Meanwhile, the metabolic pathway, the cell cycle and mitotic pathway might played vital roles in the progression of HCC.

Introduction

Hepatocellular carcinoma (HCC) is the main type of liver cancer, and causes more than 700,000 deaths each year (Ni et al., 2020; Cho et al., 2019). Recently, many studies have demonstrated that multiple genes and cellular pathways participate in the initiation and progression of HCC. The CDKN3 might play an important role in the transformation process from cirrhosis to HCC by analysis of the gene expression omnibus (GEO) data (Jiang et al., 2020). Also, miR-133a-3p might inhibit the growth of HCC by analyzing the miR-133a-3p expression and the clinicopathological characteristics of HCC based on GEO data and the Cancer Genome Atlas (TCGA) data (Liang et al., 2018). Besides, DUXAP8 might be involved in the biological processes such as cell cycle, cell division and cell proliferation in HCC, and the down-regulation of DUXAP8 inhibited the proliferation and invasion of HCC in vitro (Yue et al., 2019). Previous studies focused on the specific genes in the initiation and progression of HCC; however, the precise molecular mechanisms underlying HCC progression was not clear.

In the last decade, the high-throughput platforms were used to generate gene expression profiling in HCC. However, sequencing results are often limited and inconsistent owing to the heterogeneity of samples in independent studies, and due to the fact that most studies focus on one cohort. As such, this study sought to analyze a range of available HCC-related gene expression data sets by integrated bioinformatics analysis, with the goal of identifying potential novel hub genes and key gene module for HCC treatment and diagnosis. Currently, the Weighted Gene Co-expression Network Analysis (WGCNA), for analysis of genetic alteration during tumorigenesis, is increasingly valued as promising tools in medical oncology research. Based on WGCNA, we further explored the gene co-expression network (GCN) analysis algorithm in this study. Multiple machine learning algorithms were used to improve the reliability of the gene modules.

Besides, DNA methylation plays a role in genome stability and gene expression (Esteller & Herman, 2002). In particular, aberrant DNA promoter methylation is an important mechanism for loss of gene function in tumors (Ohtani-Fujita et al., 1993; Jarrard et al., 1998). Given that methylation is now known to play important roles in cancer, it is of great significance to detect DNA methylation of hub genes in HCC in this study.

It is vital to classify and detect the key biological pathways and hub genes participated in the initiation and progression of HCC. Firstly, we obtained the transcriptome data set of HCC and normal tissues from TCGA (Hutter & Zenklusen, 2018), and used the FC-t algorithm to identify differentially expressed genes (DEGs). Next, the Pearson correlation analysis was used to construct a GCN, and three community detection algorithms (multi-level, label-propagation, edge-betweenness) were used to identify gene modules. Then, the correlation analysis between gene modules and clinical indicators was performed, and a key gene module was identified. Herein, we performed GO/Reactome enrichment analysis on key gene module to explore the biological significance. Then, the protein protein interaction (PPI) network of the key gene module was constructed using the Search Tool for the Retrieval of Interacting Genes (STRING) database (Szklarczyk et al., 2011) , and the hub genes were identified based on PPI network. Moreover, we used the Oncomine analysis, survival analysis, GEO data set and ROC curve to verify the important roles of hub genes in HCC. Finally, we explored the methylation of hub genes using GEO data (GSE73003).

Materials and Methods

Data collection and preprocessing

A workflow of this study is shown in Fig. 1. The HCC gene expression profiles used in this study were downloaded from TCGA (https://www.cancer.gov/about-nci/organization/ccg/research/structural-genomics/tcga), which was processed using the RNA-sequencing platform, and contained 416 samples, including 367 HCC samples and 49 normal samples. In order to avoid the interference of genes with lower expression on subsequent analysis, the gene whose maximum FPKM value was less than 1 in tumor or normal samples was removed. Then, the outliers from HCC samples were removed by hierarchical clustering with R function hclust() in the stats package (v3.6.1), and euclidean distance was used in the clustering process. In this study, FC-t algorithm (Chen et al., 2018) was used to identify DEGs. The fold change of each gene’s FPKM value between cancer and normal tissues was calculated. Next, the differential expression analysis was carried out on the basis of t-test using the t.test() in stats R package (v3.6.1). Only genes with the fold change ≥ 2 or fold change ≤ 0.5 and P-value < 0.05 were regarded as DEGs.

Figure 1 Flow-chart of data analysis in this paper.

Construction of GCN and identification of gene modules

Pearson correlation analysis was used to construct a correlation matrix between pairwise DEGs with FPKM values in cancer tissues (Chang et al., 2019). And Pearson correlation analysis was implemented using the cor.test() in stats R package (v3.6.1). Then Pearson correlation coefficient —PCC— ≥ 0.65 and P-value < 0.05 was set as the cut-off criteria to screen the interaction between two genes. The reserved interactions were represented by networks, the largest of which was the GCN. We clustered all genes (nodes) in the GCN with three different community detection algorithms, including multi-level (Blondel et al., 2008), label-progration (Raghavan, Albert & Kumara, 2007) and edge-betweenness (Newman & Girvan, 2004), which were performed by R functions multilevel.community(), label.propagation.community(), edge.betweenness.community() in the igraph package (v1.2.4) (Csardi & Nepusz, 2006). Then the modularity was used to evaluate the clustering results, so as to select the optimal module identification result.

Association analysis between gene modules and clinical indicators

Principal component analysis (PCA) (Wold, Esbensen & Geladi, 1987) was used to analyze the gene expression profiles in each module by using the prcomp() in stats R package (v3.6.1). Then the first principal component was defined as the module eigengenes (MEs). We believed that the four clinical indicators of event, T, N, and M (T referred to the primary tumor stage, N referred to the regional lymph node involvement stage and M referred to the distant metastasis stage) were of great significance for judging the initiation and progression of tumors, so an association matrix was constructed based on the correlation analysis between the MEs and event, T, N and M, which was calculated by the Pearson correlation analysis. Then, the gene module which was highly related to clinical indicators was selected as the key gene module.

To further verify the importance of the key module, Cox proportional hazards regression model was used to perform survival analysis on genes in all modules, and P-values of Cox regression results were obtained (the survival analysis results came from the online analysis tool onclnc (http://www.oncolnc.org/)). Further, the prognostic significance (PS-value, calculated in logs as -lgP-value) was used to measure the importance of a gene, and the PS-value of a module is the sum of the PS-values of all genes in this module. Obviously, the larger the PS-value of a module, the more important the module is.

GO/Reactome enrichment analysis

For the biological significance of the key gene module, the genes in which were enriched with the biological processes provided by the GO database (http://geneontology.org/) and the signaling pathways provided by the Reactome database (https://reactome.org/). GO enrichment analysis was implemented with the enrichGO() in clusterProfiler R package (v3.12.0), and Reactome enrichment analysis was performed by logging in the online database, P-value < 0.05 was considered statistically significant. Meanwhile, the top 20 GO terms and signaling pathways with the lowest P-value were selected for further research. Finally, the GO terms were categorized with QucikGO database (https://www.ebi.ac.uk/QuickGO/).

Construction of PPI network and analysis of hub genes

The online database STRING (https://string-db.org/) was applied to construct a PPI network of the genes in key gene module and analyze the functional interactions between proteins. A confidence score ≥ 0.400 was set as significant. Subsequently, the result was visualized using Cytoscape software (v3.7.1) (Shannon et al., 2003), and nine topology analysis algorithms (DNMC, MNC, Degree, EPC, BottleNeck, Closeness, Radiality, Betweenness, Stress) provided by the cytohubba (Chin et al., 2014) plug-in were used to calculate the importance of nodes (genes) in the PPI network. For details of the nine topology analysis algorithms, please refer to the literature (Chin et al., 2014). Further, five genes with the highest score in each algorithm were merged together as the hub genes.

Validation of hub genes

Firstly, the mRNA expression of hub genes was explored in common cancer using Oncomine database (https://www.oncomine.org). The parameters were set as follows: THRESHOLD (P-VALUE) = 0.05, THRESHOLD (FOLD CHANGE) = 2. Then the online analysis tool onclnc was used for the survival analysis of hub genes. And the cancer was set as LIHC, lower percentile was set as 20, and upper percentile was set as 20. Furthermore, we downloaded a test data set, GSE138485, from the GEO (https://www.ncbi.nlm.nih.gov/geo/), and this data set included 64 paired normal and HCC samples (Table S1). The t-test was used to verify the differential expression of the hub gene in GSE138485. Moreover, ROC curve and AUC were used to detect the ability of hub genes to distinguish tumors from normal tissues.

DNA methylation analysis

The gene methylation profiling data set GSE73003 was downloaded from the GEO (Table S2). It included 40 paired normal and HCC samples from 20 patients. We found the methylation changes of the hub genes in GSE73003, and then used t-test to identify the genes whose methylation changed significantly.

Results

Identification of DEGs in expression profiles

The HCC gene expression profiles contained 416 samples, including 367 HCC and 49 normal samples, and the original HT-Seq-FPKM data included 60,483 genes in total. The genes with lower expression were removed. Then, the remained 14,129 genes (Table S3) were used for hierarchical clustering to obtain a data set for further analysis. The result showed that there were three outlier samples should be removed, TCGA-DD-AAEB, TCGA-CC-5259 and TCGA-FV-A4ZP (Fig. S1). Using bioinformatics approaches, a total of 4,601 DEGs between HCC and normal samples were identified, including 4,130 up-regulated and 471 down-regulated genes (Fig. 2, Table S4).

Figure 2 Identification results of DEGs.

(A) X-axis represents log2 fold-changes and Y-axis represents negative logarithm to the base 10 of the P-values. Black vertical and horizontal dashed lines reflect filtering criteria (log2(Fold change) = ±1 and P-value = 0.05). (B) Pink and green bars are number of significantly up-regulated (n = 4,130) and down-regulated genes (n = 471) in HCC compared with its normal tissues.

Construction of GCN and identification of gene modules

Pearson correlation analysis was used to construct a correlation matrix with FPKM values between pairwise DEGs. In total, there were 21,169,201 interactions. After filtered, 2,859 genes and 57,340 interactions were kept and imported into Cytoscape software for visualization. In total, 95 networks were built (Fig. 3). The result showed that there were 2,583 genes in the large network, while there were fewer than 20 genes in each smaller one. After removed the small networks, the largest one which referred to GCN were kept for the further research.

Figure 3 The GCN was constructed by Pearson correlation analysis.

The total number of gene in figure is 2,859, and the number of gene in GCN (the largest network) is 2,583.

To obtain more accurate and objective clustering results, three community discovery algorithms were used to cluster all genes (nodes) in the GCN. The modularity of each algorithm was shown in Table 1, and the modularity of multi-level was 0.6009015, label-propagation was 0.3748268 and edge-betweenness was 0.4815381, respectively. The multi-level algorithm clustering result with the highest modularity was performed for subsequent analysis. A total of nine modules were identified after removing the modules with less than 50 genes (Fig. 4, Table S5). The network density of these nine modules was shown in Table 2; the density of m9 was the lowest one, 0.065630124. It was worth noting that the network density of these nine modules was greater than that of GCN (0.01704045).

Table 1 The modularities of 3 algorithms.

Algorithm	Modularity	
multi-level	0.6009015	
label-propagation	0.3748268	
edge-betweenness	0.4815381	

Figure 4 Module identification result obtained by the multi-level algorithm.

The multi-level algorithm divided GCN into 13 gene modules, the little modules with less than 50 genes were removed, and the labeled modules were nine gene modules for further analysis.

Identification of key gene module and GO/Reactome enrichment analysis

The first principal component of PCA result which performed on the gene expression profiles in each module defined as the MEs (Table S6). Furthermore, an association matrix was constructed based on the correlation analysis between the MEs and the clinical indicators, including event, T, N and M, which was calculated by the Pearson correlation analysis. In this process, a key gene module m1 was obtained from the association matrix, and the correlation coefficient between m1 and T was the maximum 0.259, m1 and event was 0.179, as well as m1 and N was 0.080 (Fig. 5). In addition, we calculated the PS-values of all modules, and the PS-value of module m1 was the largest (Fig. 6).

Table 2 The network densities of nine gene modules containing more than 50 genes.

Module	Densities	
m1	0.197628458	
m2	0.29029703	
m3	0.177997842	
m4	0.215233881	
m5	0.106922766	
m6	0.090001227	
m7	0.080307853	
m8	0.216806723	
m9	0.065630124	

To further investigate the function of identified genes in m1, GO enrichment analysis was performed to analyze functional enrichment (Table S7). The top 20 biological processes enriched in GO terms were shown in Fig. 7. The genes in m1 mainly participated in biological processes associated with the process of mitosis and meiosis, and the functions of catalytic and exodeoxyribonuclease activity. Moreover, the biological processes of key gene module mostly occurred in the chromosomal region. Additionally, the genes in m1 were enriched in many Reactome signaling pathways (Table S8), which mainly included the cell cycle and mitotic (Table 3).

Figure 5 The heat map of the correlation between gene modules and clinical indicators.

The row corresponds to module, and the column corresponds to clinical indicator. The m1 is key gene module in HCC.

Figure 6 The PS-values of all modules.

Figure 7 The 20 GO Terms (biological processes) with the smallest P-value of genes in the key gene module (m1).

The size of bubbles represents the numbers of genes, the color of bubbles corresponds to P-value, and the GeneRatio represents the ratio of the number of genes enriched to the total number of genes in key gene module (m1).

Table 3 The 20 signaling pathways with the smallest P-value of genes in the key gene module (m1).

ID	Description	Count	P-value	
R-HSA-69190	DNA strand elongation	18	1.11E−16	
R-HSA-453279	Mitotic G1 phase and G1/S transition	31	1.11E−16	
R-HSA-69278	Cell Cycle, Mitotic	78	1.11E−16	
R-HSA-1640170	Cell Cycle	90	1.11E−16	
R-HSA-73894	DNA Repair	38	1.11E−16	
R-HSA-69620	Cell Cycle Checkpoints	35	1.11E−16	
R-HSA-69242	S Phase	27	2.22E−16	
R-HSA-69206	G1/S Transition	25	3.33E−16	
R-HSA-69306	DNA Replication	24	8.88E−16	
R-HSA-69239	Synthesis of DNA	23	2.33E−15	
R-HSA-68886	M Phase	36	8.33E−15	
R-HSA-73886	Chromosome Maintenance	20	3.18E−14	
R-HSA-5693532	DNA Double-Strand Break Repair	24	3.45E−14	
R-HSA-5693538	Homology Directed Repair	21	1.63E−13	
R-HSA-68877	Mitotic Prometaphase	24	4.54E−12	
R-HSA-69186	Lagging Strand Synthesis	11	5.40E−12	
R-HSA-5693567	HDR through Homologous Recombination (HRR) or Single Strand Annealing (SSA)	19	5.88E−12	
R-HSA-73933	Resolution of Abasic Sites (AP sites)	13	1.53E−11	
R-HSA-176187	Activation of ATR in response to replication stress	12	2.21E−11	
R-HSA-4615885	SUMOylation of DNA replication proteins	13	2.52E−11	

Figure 8 The PPI network of key gene module (m1).

Each node corresponds to a gene, where a red node corresponds to a hub gene.

Construction of PPI network and identification of hub genes

Based on the STRING database, a PPI network for all genes in the m1 was constructed (Fig. 8). Nine topological analysis algorithms provided by cytohubba plug-in were used to calculate the importance of node (gene) in the PPI network (Table S9). The five genes with the highest score in each algorithm were merged together to be the hub genes in this study. Finally, 16 hub genes were identified, NUSAP1, MCM3, TRMT6, RFC3, POLA2, AURKA, CDC20, TOP2A, ECT2, TK1, MCM2, FEN1, NOP58, GINS2, NCAPD2 and KPNA2 (Table 4). It was worth noting that the 16 hub genes were all up-regulated genes.

Validation of hub genes

Firstly, the mRNA expression of 16 hub genes in liver cancer was explored using Oncomine analysis. The result showed that 13 hub genes were up-regulated in liver cancer as shown in Fig. 9. Then we found that the expression levels of 13 hub genes were significantly related with worse overall survival (OS) (Logrank P-value < 0.05) (Fig. 10). After the merger, a total of 11 genes meet the above two requirements, which included MCM3, TRMT6, AURKA, CDC20, TOP2A, ECT2, TK1, MCM2, FEN1, NCAPD2 and KPNA2. And the following focuses on these 11 hub genes.

Further, the data of GEO (GSE138485) showed that the RPKM of 11 hub genes were significantly (all P-values < 0.005) up-regulated in HCC samples compared with normal samples (Fig. 11). Moreover, based on the RPKM of 11 hub genes in the GEO data set, we used ROC curve and AUC to classify HCC and normal samples. The results showed that the whole 11 hub genes had highly diagnostic efficiencies to distinguish tumors from normal tissues (AUC > 0.87 and P-value < 1.0E−06) (Fig. 12).

Table 4 Calculation results of nine algorithms.

The row corresponds to gene, and the column corresponds to algorithm. Each value represents a score.

Hub gene	DMNC	MNC	Degree	EPC	BottleNeck	Closeness	Radiality	Betweenness	Stress	
NUSAP1	1.0893	51	51	19.988	9	101.33333	5.09091	96.90787	1902	
MCM3	0.91716	69	69	22.824	23	111.5	5.24242	372.309	7082	
TRMT6	0.47733	9	11	2.093	6	69.78333	4.4	1264.09432	10640	
RFC3	0.88611	60	60	20.772	10	107.25	5.19394	542.48026	7874	
POLA2	1.16211	36	36	16.597	1	93.33333	4.97576	27.46845	786	
AURKA	0.83591	71	71	23.302	7	112.5	5.25455	944.81247	9646	
CDC20	0.75746	80	80	23.462	5	117.66667	5.33939	1556.30666	16426	
TOP2A	0.81632	76	76	23.363	2	115.66667	5.30909	1108.53404	13048	
ECT2	1.16516	30	30	14.084	1	89.33333	4.90303	9.41338	276	
TK1	1.14366	41	41	17.127	1	95.75	5	60.29409	1056	
MCM2	0.93961	69	69	22.807	5	111.58333	5.25455	827.49161	8772	
FEN1	0.82386	73	73	22.298	4	114.25	5.29697	1278.36825	12488	
NOP58	0.2704	15	16	2.45	10	76.91667	4.61818	2059.91502	14078	
GINS2	1.15247	50	50	19.731	1	101	5.09091	105.79312	1852	
NCAPD2	1.19618	38	39	17.413	7	96.25	5.05455	1216.19414	12076	
KPNA2	0.97174	51	51	19.066	8	104.41667	5.21212	1504.41719	14138	

Figure 9 The results returned from Oncomine database.

The row corresponds to cancer, and the column corresponds to gene. The red square represents that the gene was up-regulated in cancer, the blue square represents that the gene was down-regulated in cancer, and the value in the square represents the number of related references.

Figure 10 Significant correlation between hub genes expression and survival.

Survival curves of genes (A) AURKA, (B) CDC20, (C) ECT2, (D) FENI, (E) GINS2, (F) KPNA2, (G) MCM2, (H) MCM3, (I) NCAPD2, (J), NOP58, (K) NUSAP1, (L) POLA2, (M) RFC3, (N) TK1, (O) TOP2A, (P) TRMT6. X-axis represents survival time and Y-axis represents survival rate.

Figure 11 Te heat map of RPKM of KPNA2, MCM3, FEN1, TRMT6, and others. in normal and HCC samples.

TA-Tf represents HCC samples, NTA-NTf represents normal samples.

Figure 12 The ROC curves of 11 hub genes.

(A) AURKA, (B) CDC20, (C) ECT2, (D) FENI, (E) KPNA2, (F) MCM2, (G) MCM3, (H) NCAP2, (I) TK1, (J), TOP2A, (K) TRMT6. These ROC curves described the diagnostic efficiency of the mRNA levels of 11 hub genes for HCC and normal tissues.

DNA methylation analysis of the hub genes

Among the 11 hub genes, we found the methylation of gene CDC20, TOP2A, TK1, FEN1 were significantly changed in the gene methylation profiling data set (GSE73003) (P-value < 0.05), and they were hypomethylation in the HCC samples (Table 5). It was noted that the expression (RPKM) of CDC20, TOP2A, TK1, FEN1 were significantly higher in HCC samples compared to normal tissues.

Table 5 Methylation of hub genes.

The P-value was obtained by t-test. Methylation status was obtained by analyzing GSE73003, and expression status was obtained by analyzing the TCGA data set.

Hub gene	Methylation status	P-value	Expression status	P-value	
CDC20	Hypomethylation	8.94E−05	High expression	2.11E−32	
TOP2A	Hypomethylation	4.41E−02	High expression	2.70E−41	
TK1	Hypomethylation	4.78E−04	High expression	1.02E−49	
FEN1	Hypomethylation	4.06E−02	High expression	8.23E−62	

Discussions

On the global scale, HCC is a major contributor to both cancer incidence and mortality. Understanding the molecular mechanism of HCC is of critical importance for early detection, diagnosis, and treatment. In our study, the HCC mechanism was analyzed by bioinformatics analysis, including DEGs screening, GCN construction, module analysis, hub gene identification in the PPI network, validation of the hub genes, and DNA methylation analysis of the hub genes. These findings may help us to understand the molecular mechanism of HCC pathogenesis and identify potential biomarkers for the diagnosis and treatment of HCC.

From the result of module identification, we found that the network density of each module was greater than that of GCN. It might imply that compare with the other genes in the GCN, the genes in one module perform the same biological function, which also proved the reliability of the multi-level algorithm.

GO enrichment analyses showed that the key gene module were associated with many biological processes. Previous reports showed that the tRNA expressed abnormal (GO:0006409, GO:00071431, GO:0051031 and GO:0006403) had a dual role for the promotion and suppression in cancer development (Nientiedt et al., 2016). The tRNA might be involved in cell proliferation process, cell cycle and gene regulation process in cancer (Balatti et al., 2017; Goodarzi et al., 2015). In tumor cells, the variation of cell function was often influenced by the structure of DNA strand and the conformation of chromosome (GO:0009987 and GO:0071103). DNA was regulated by different functional regions in the nucleus due to its local strand structure abnormality, compression or long-range proximity (Taberlay et al., 2016). Besides, signal transduction by p53 class mediator (GO:0072331) and response to heat (GO:0009408) belonged to response to stimulus (GO:0050896). P53 is a key tumor suppressor (Vogelstein, Lane & Levine, 2000). As a transcription factor, p53 transcribes its target genes to regulate various cellular biological processes, including cell cycle arrest, apoptosis, senescence, energy metabolism, and anti-oxidant defense, to prevent tumorigenesis (Feng & Levine, 2010). P53 is an important tumor suppressor gene, 30 to 60% of HCC patients with mutated p53 gene (Hussain et al., 2007). Many GO Terms belonged to cellular process (GO:0009987), DNA replication (GO:00006260) and protein sumoylation (GO:0016925). The post-translational modification sumoylation is a major regulator of protein function that plays an important role in a wide range of cellular processes (GO:0009987 and GO:0016925) (Wilkinson & Henley, 2010). Furthermore, GO enrichment analyses results showed that the small networks were associated with the detoxification of inorganic compound (GO:0061687), nucleosome assembly and organization (GO:0006334, GO:0034728), metabolism, such as fatty, hormone and neurotransmitter. Similarly, the small gene modules were associated with antigen processing and presentation (GO:0019882), complement activation lectin pathway (GO:0001867), RNA splicing (GO:0008380), negative regulation of lymphocyte (GO:0050672) and centromeric sister chromtid (GO:0070601). Beyond these, previous reports revealed the evidence of meiosis (GO:0140014 and GO:1901990) might be responsible for the proliferation of the tumor cells, such as meiosis error invoked the malignant transformation of germ cell tumor (Ichikawa et al., 2013). Besides, the studies demonstrated that oocyte meiosis might induce the proliferation of the tumor cells (Li et al., 2014). It has been reported that increased expression genes which were associated with cell cycle and oocyte meiosis, were associated with the development and progression of HCC (Fujii et al., 2006; Zhang et al., 2019).

According to the validation, the 11 hub genes were good biomarkers in HCC and functioned as tumor promoter. MCM3 complex required for cell cycle regulation of DNA replication in vertebrate cells (Madine et al., 1995). MCM3 were significantly up-regulated in invasive ductal carcinoma (Zhao et al., 2020). Consistent with the findings reported in previous studies (Zhuang, Yang & Meng, 2018; Yang, Pan & You, 2019), our results showed that higher MCM3 expression levels are associated with worse clinical outcomes and a shorter survival times of patients with HCC, thereby highlighting the potential use of MCM3 as a prognostic biomarker. MCM2 is a promising marker for premalignant lesions of the lung (Yan, Merchant & Tye, 1993; Musahl et al., 1998). Besides, high MCM2 expression has been associated with poor prognosis in HCC patients (Liu et al., 2018; Li et al., 2019). Previous report showed that the up-regulation of Epithelial cell transforming sequence 2 (ECT2) was significantly associated with early recurrent HCC disease and poor survival. ECT2 was closely associated with the activation of the Rho/ERK signaling axis to promote early HCC recurrence. Moreover, knockdown of ECT2 markedly suppressed Rho GTPases activities, enhanced apoptosis, attenuated oncogenicity and reduced the metastatic ability of HCC cells (Chen et al., 2015). The NCAPD2 is a novel candidate genes in ovarian cancer (Tatsumoto & T, 1999; Fields & Justilien, 2009). In fact, elevated AURKA expression was observed in several human cancers, such as pancreatic cancer, endometrioid ovarian carcinoma and colorectal cancer liver metastasis, and was associated with poor prognosis (Furukawa et al., 2006). Besides, AURKA promoted cancer metastasis by regulating epithelial-mesenchymal transition and cancer stem cell properties in HCC (Chen et al., 2017). Emerging evidence suggests that KPNA2 plays a crucial role in oncogenesis and early recurrence. KPNA2 expression levels were found to be markedly higher in tumor tissue (83.33%, 25/30) (Feng et al., 2014). Besides, nuclear KPNA2 expression was significantly up-regulated (30.3%, 67/221) in HCC tissues; however, no nuclear expression of KPNA2 in non-tumorous tissues was observed by immunohistochemical assays (Jiang et al., 2014). Both in vitro and in vivo experiments demonstrated that knockdown of KPNA2 reduced migration and proliferation capacities of HCC cells, while over-expression of KPNA2 increased these malignant characteristics (Xinggang et al., 2019). TOP2A encodes a 170 kDa nuclear enzyme that controls DNA topological structure, chromosome segregation, and cell cycle progression (Isaacs et al., 1998). TOP2A over-expression, that is, increased level of TOP2A mRNA and protein, has been detected in HCC (Panvichian et al., 2015; Wong et al., 2009). TRMT6, which catalyzes the installation of m1A at position 58 of tRNA, is an oncogene in HCC (Li et al., 2017). Clinical significance of TRMT6 in HCC and colon cancers is very important (Wang et al., 2019). TRMT6 knockout HCC cells displayed compromised stemness properties, as reflected by impaired sphere formation and tumor initiating ability, and increased sensitivity to molecular target drug sorafenib (Chen, 2019). TRMT6 was up-regulated in HCC tissues, and higher TRMT6 expression levels was correlated with reduced OS (P = 0.0224) and RFS (P = 0.0146) in patients with primary HCC (Wang et al., 2019). The above results indicated that TRMT6 might be a promising prognostic biomarker for poor clinical outcomes in primary HCC patients.

Previous results showed that the levels of serum TK1 (thymidine kinase 1) in the primary hepatic carcinoma group were significantly higher than those in the control group and the benign group (P ≤ 0.05) (Shen-Jie & Li, 2018; Zhang, Lin & Li, 2015). Cell division cycle 20 (CDC20) encodes a regulatory protein interacting with the anaphase-promoting complex/cyclosome in the cell cycle and plays important roles in tumorigenesis and progression of multiple tumors (Liu et al., 2015). Immunohistochemistry result showed that, in the 132 matched HCC tissues, high expression levels of CDC20 were detected in 68.18% HCC samples, and over-expression of CDC20 was positively correlated with gender (P = 0.013), tumor differentiation (P = 0.000), TNM stage (P = 0.012), P53 and Ki-67 expression (P = 0.023 and P = 0.007, respectively) (Li et al., 2014). The Flap endonuclease (FEN1) expression levels were also positively correlated with tumor size (P = 0.047 < 0.05), distant metastasis (P = 0.013 < 0.05) and vascular invasion (P = 0.024 < 0.05) in HCC (Li et al., 2019). Combined with the study in this paper, it was reasonable to speculate that these 11 hub genes might be biomarkers for HCC.

DNA methylation, a pretranscriptional modification, regulates the stability of gene expression states and maintains genome integrity by collaborating with proteins that modify nucleosomes (Jaenisch & Bird, 2003; Zhong, Agha & Baccarelli, 2016). Altered DNA methylation such as tumor suppressor gene hypermethylation or oncogene hypomethylation is thought to promote tumorigenesis (Ehrlich, 2019). Previous reports found that genes including P15, P16, RASSF1A and Retinoblastoma 1 were inactivated in HCC due to promoter hypermethylation of these genes (Fan et al., 2018). In the present study, we identified four highly-expressed hub genes with hypomethylation, CDC20, TOP2A, TK1, FEN1. Therefore, we might provide more effective diagnostic strategies by these novel biomarkers of HCC.

In accordance with our findings, previous studies have also identified hub genes that participate in HCC (Hua et al., 2020; Song et al., 2020). Shengni et al. constructed a PPI network based on 176 DEGs. Then four gene modules were identified with Cytoscape MCODE plug-in and 12 hub genes were obtained with integrated survival and methylation analysis. Finally, three genes, KPNA2, TARBP1 and RNASEH2A, were identified as diagnostic and prognostic markers for HCC. However, 176 genes were a little too small to construct a network and identify gene modules, and all of DEGs were up-regulated. It would be better if this article had an association analysis between gene modules and clinical indicators. Another study identified key gene modules with WGCNA. And 29 hub genes were identified based on PPI network. Finally, DAO, PCK2, and HAO1 were determined as prognostic targets for HCC. It was worth noting that the study used Pearson correlation analysis to associate gene modules with clinical indicators, which was classic but a little simple. Besides, they used the ”Degree” algorithm to identify hub genes in the PPI network, rather than make full use of the topology of the PPI network. In our study, data analysis was conducted using the integrated bioinformatics analysis based on GCN analysis, which is highly suitable for the analysis of gene expression data. First of all, 4,601 DEGs (including 4,130 up-regulated and 471 down-regulated genes) were used to construct GCN. In particular, we used three community detection algorithms to identify gene modules, and used modularity to select the optimal module identification result. In addition, Pearson correlation analysis and survival analysis were used to identify key gene module, so the key gene module of HCC in this paper was more accurate, and GO enrichment analysis results also proved the reliability of this module. Lastly, nine topology analysis algorithms were used to identify hub genes in PPI network. And we used the Oncomine analysis, survival analysis, GEO data set and ROC curve to verify the important roles of hub genes in HCC. Therefore, the hub genes identified in the present study are more reliable and comprehensive.

Conclusions

In summary, integrated bioinformatics analysis based on GCN analysis was built to analyze the gene expression profile of HCC, and the hub genes and biological pathways in HCC were identified in this study. MCM3, TRMT6, AURKA, CDC20, TOP2A, ECT2, TK1, MCM2, FEN1, NCAPD2 and KPNA2 could be potential biomarkers or therapeutic targets for HCC. Meanwhile, the metabolic pathway, the cell cycle and mitotic pathway might played vital roles in the progression of HCC. However, we need more experiments to investigate these novel, key and hub genes. Based on these results, the underlying molecular mechanisms of HCC were explored on genetic and molecular levels, which provided new insights into HCC diagnosis and treatment.

Supplemental Information

Supplemental Information 1 Hierarchical clustering tree of tumor tissue samples

Sample clustering was conducted to detect outliers, whlie TCGA-DD-AAEB, TCGA-CC-5259 and TCGA-FV-A4ZP were removed.

Click here for additional data file.

Supplemental Information 2 The test dataset GSE138485

TA-Tf represents HCC samples, NTA-NTf represents normal samples.

Click here for additional data file.

Supplemental Information 3 The gene methylation profling dataset GSE73003

TargetID is the number of the methylation site, and GeneName represents the gene corresponding to this site. TA-TT represents HCC samples, NTA-NTT represents normal samples.

Click here for additional data file.

Supplemental Information 4 Preprocessed data for DEGs identification

Click here for additional data file.

Supplemental Information 5 Calculation results of FC-t algorithm

The results which include the Fold change and P-value (t-test) of each gene.

Click here for additional data file.

Supplemental Information 6 Details of 9 modules

The results which show the genes contained in each module.

Click here for additional data file.

Supplemental Information 7 The ME of each module

Click here for additional data file.

Supplemental Information 8 GO enrichment results of key gene modules

Gene list represents the list of genes enriched in this Term, and Count represents the number of genes enriched in this Term.

Click here for additional data file.

Supplemental Information 9 Reactome. enrichment results of key gene modules

Gene list represents the list of genes enriched in this Term, and Count represents the number of genes enriched in this Term.

Click here for additional data file.

Supplemental Information 10 Hub genes identified by 9 topological analysis

Including scores from 9 topological analysis algorithms for each gene in PPI network. The row corresponds to gene, and the column corresponds to algorithm. Each value represents a score.

Click here for additional data file.

Additional Information and Declarations

Competing Interests

Author Contributions

Data Availability

The authors declare there are no competing interests.

Qian Zhao performed the experiments, analyzed the data, authored or reviewed drafts of the paper, and approved the final draft.

Yan Zhang performed the experiments, analyzed the data, prepared figures and/or tables, and approved the final draft.

Shichun Shao performed the experiments, prepared figures and/or tables, and approved the final draft.

Yeqing Sun conceived and designed the experiments, authored or reviewed drafts of the paper, and approved the final draft.

Zhengkui Lin conceived and designed the experiments, prepared figures and/or tables, and approved the final draft.

The following information was supplied regarding data availability:

Data is available at GEO (GSE138485, GSE73003) and in the Supplementary Files.

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
