# Peer review of "Identification of hub genes and biological pathways in hepatocellular carcinoma by integrated bioinformatics analysis"

_PeerJ, doi:10.7717/peerj.10594_

## Round 0.1 · original submission · Major Revisions

You will see that, while the reviewers find your work of interest, they have raised points that need to be addressed before we can make a decision on publication.

Reviewer 1 ·

Basic reporting

1. Several figures, such as Figure1 and Figure7, was unclear to understanding.
2. Typing errors or ambiguous words, like line 164 “functionally significant”?

Experimental design

Methods and data resources lacks detailed information.
1. What does adjacent tissues mean? Why you define them as adjacent samples?
2. What is the cutoff of a gene with lower expression? And why you choose the rule?
3.The values of PCC (Pearson correlation coefficient) can be -1 to 1. I think the 0.65 cutoff is roughly without any reference or statistical explanation.
4. The description of your association matrix was too simple. How to build it and what’s the calculation formula?
5. The method of hierarchical clustering is unclear, is it average correlation value or others?
6. The legend of Figure 2 was rong. It should be log2(Fold
Change)=±1
7.In Figure 3, there are several networks. The author only used the largest one. They did not discuss why they are smaller and the possible function of the smaller networks.
8. The details of modules were unclear. How many genes were included in the 9 modules? And why to remove modules with genes less than 50?
9. In Line 158, the two BP entries were not found in figure 6.
10. The heatmap of Figure 10 should be sorted or clustered.

Validity of the findings

1. Due to lack of data resources, it is uncertain if the test samples(GSE138485) were involved in the tissues samples of network construction.
2. In the validation section of 13 hub genes, several genes are up-regulated and down-regulated together in the same cancer, why?
3. The Receiver Operating Characteristic Curve (ROC) and AUC value are suggest to validate the function of 11 hub genes.
4. The analysis of DNA methylation is too simple. There are several kinds of DNA methylation,so which methylation was analyzed here?
5. In line 274-276, why you mentioned the lncRNA, is it correlated with your results?
6. In the discussion section, there is too more introduction of GO entries, and lacks validations to confirm the hypothesis that those hub genes are biomarkers of HCC.

Reviewer 2 ·

Basic reporting

The English language should be improved. Some examples include lines 82, 149, 165, etc.
In lines 82 and 149, “a association matrix” should be “an association matrix”.
In lines 165, “9 topological analysis algorithms” should be “Nine topological analysis algorithms”; it is better to use English words instead of Arabic numerals less than ten.

The logic of the introduction needs to be improved. The introductory part should provide an analytical framework for the current research. Relevant prior literature should be appropriately referenced.

In lines 57-59, the details of the dataset(s) and samples should be provided.
In line 66, P-value ≤ 0.05 was used as significant, while in other parts of the manuscript P<0.05 was chosen. It should be better to unify the standard.
In lines 100-104, the details of the 9 topology analysis algorithms should be provided.
In Figure 2, it seems that down-regulated (n=471) and up-regulated genes (n=4,130) were reversed; Additionally, bar colors seem to be pink and green, not red and blue.

The results and discussions are not sufficient. Need to discuss the causality or relationship of modules, gene functions, pathways and HCC stages.

Experimental design

no comment

Validity of the findings

Impact and novelty not assessed.

The data in the results section should be displayed more to ensure that the results are credible rather than statistically false positives. For example, in lines 133-137 & 142-143, since there are so many genes and interactions, does the screening criterion only depend on the size of the subnet? As far as we know, the greater the amount of data, the greater the power (detection effectiveness); In addition, the occurrence of cancer may just be some mutation(s) of a single gene or genes related to its pathway, so small subnets with special functions also need attention. Another example, in subsection 3.3 (lines 147-161), the module m1 has a high risk of false positive. It is hard to explain why m1 was associated with meiosis. Since the occurrence and progression of tumors is a multi-step process, a more reasonable approach is to subdivide the modules and correspond to different stages. Supplementary materials need to add necessary notes.

In the present study, the analysis framework seems not clear enough. The purpose and function of each step of analysis need to be clearer. For example, in lines 59-60, why genes with lower expression should be removed? In lines 82-86, why the clinical indicators were divided into event, T, N and M stages? Moreover, Pearson correlation analysis is too simple for the association analysis between gene modules and clinical indicators. Time series analysis model, such as survival analysis should be conducted. In lines 99-105, what is the basis or reference for “confidence score ≥ 0.400 was set as significant” or “5 genes with the highest score in each algorithm were merged together as the hub genes”? In lines 114-115, parameter settings for random forest(s) need to be provided.

Conclusions are not well stated.

Additional comments

Sufficient research should be done before starting the study. Previous literature(s) related to this study should be cited and the results should be compared with the present study.

---

## Round 0.2 · Minor Revisions

Please address the reviewers's comments by including more detailed description in the figures and the content.

Reviewer 1 ·

Basic reporting

no comment.

Experimental design

1. In figure3, the total number of gene and the number of gene in GCN should be listed. How much gene were filtered based on the smaller networks?

2. The author finally identified 9 modules. How to define a module is “too few/outliers” and a module is “too large”? Is there any detailed parameters ?

3. In the validation section of 13 hub genes, author explained several genes up-regulated and down-regulated together were due to inconsistent data. Thus, what is the problem make them inconsistent? Did you make quality control before data processing? Are these data sets credible?

4. There are three types of DNA methylation, CHH, CHG and CG. Which kind did you analyze here?

Validity of the findings

no comment.

Reviewer 2 ·

Basic reporting

Supplementary tables need to be sorted logically, for example, in ascending order of P value. In addition, some headers need to be annotated.
There is still some spelling or grammar errors, such as "mutilevel" in line 198.

Experimental design

no comment

Validity of the findings

no comment

Additional comments

Regarding the meiosis in the m1 module, it is better to provide some explanation in the discussion section. There are several articles on meiosis and tumors.

---

## Round 0.3 · Minor Revisions

Recently, a few similar studies has been published, example include:
https://doi.org/10.3389/fgene.2020.00895
https://www.aging-us.com/full/12/5439

The method and conclusions present in this study has overlaps with the studies listed above, but the authors failed to cite these paper. I suggest to make comparison with these studies and demonstrate what is the added value for this study.

---

## Round 0.4 · accepted · Accept

I appreciate the authors' response to my comment. The revised version is acceptable.